# The Development of a Simple Projection-Based, Portable Olfactory Display Device

**DOI:** 10.3390/s23115189

**Published:** 2023-05-30

**Authors:** Chuhong Wang, James A. Covington

**Affiliations:** School of Engineering, University of Warwick, Coventry CV4 7AL, UK; j.a.covington@warwick.ac.uk

**Keywords:** aroma generator, olfactory display, vortex ring, smell generation

## Abstract

Olfactory displays are digital devices designed to provide the controlled release of odours to users. In this paper, we report on the design and development of a simple vortex-based olfactory display for a single user. By employing a vortex approach, we are able to minimize the amount of required odour, whilst still producing a good user experience. The olfactory display designed here is based on a steel tube with 3D-printed apertures and solenoid valve operation. A number of different design parameters (such as aperture size) were investigated, and the best combination was combined into a functional olfactory display. User testing was undertaken with four volunteers who were presented with four different odours, at two concentrations. It was found that the time to identify an odour was not strongly related to concentration. However, the intensity of the odour was correlated. We also found that there was a wide variance in human panel results when considering the length of time for a subject to identify an odour to its perceived intensity. This is likely linked to the subject group receiving no odour training before the experiments. However, we were able to produce a working olfactory display, based on a scent project method, which could be applicable to a range of application scenarios.

## 1. Introduction

Research into digital devices for odour generation, or so-called ‘olfactory displays’, remains relatively limited compared with technologies developed to replicate other human senses. This is evident from the very small number of publications compared to those associated with, for example, sight and sound. However, the number of potential applications is considerable and as diverse as education [1], entertainment [2] and even medicine [3]. For example, olfactory displays have been used in the treatment of war veterans with post-traumatic stress disorder (PTSD) [4] as well as in simpler applications, such as wine aroma training [5]. Furthermore, recent research has shown that olfactory displays can greatly enhance user experience during gaming [6]. Despite these discoveries, research into olfactory displays is still limited. The reason behind this is complicated, but may include the lack of a convenient olfactory device, difficulties with scent generation/removal and unsuccessful previous attempts at commercialization. One example of such a company is Feelreal, which promised a multisensory mask for VR gaming, but has currently failed to deliver a commercial product [7].

There have been a small number of prototypes and commercial devices over the past 60 years. These can be divided into two types: ubiquitous or wearable [8]. Ubiquitous type products function by releasing the scent into a closed area where the user (or users) are located, via methods such as an electrical fan [9], piezoelectric [10] or a heater [11]. This type of device is similar to a digital scent diffuser. The issue with this approach is that over time the amount of scent particles in an area gradually builds up—reducing the overall user effect. This not only results in a need for extensive ventilation, but also means that other people in the same location will smell the scent even though they are not using the device. Therefore, it is less applicable in a consumer setting (such as in the home) though it could be used in cinemas or theme parks where everyone is enjoying the same entertainment. Alternatively, wearable type devices are worn by individual users in operation. One example is InScent, an olfactory device designed to be a pendant worn by the user [12]. The product is designed so that when a digital message is received, a certain scent will be released. This has the same problem as the ubiquitous products as the scent potentially can reach everyone in the same room. This is a challenge that many olfactory display devices face. Therefore, for a consumer setting, it is best to only release a minimal number of scent particles to achieve the desired effect. This is particularly relevant for VR and gaming scenarios. Recently, one group has developed an olfactory display for virtual reality glasses. In their paper, they described a low-cost wearable olfactory display that could be put behind a Google Cardboard for an effective scent display [13]. A group of 32 volunteers were recruited to test this device alongside a VR game. Some positive feedback indicated that olfactory displays can make users feel more immersed in the visual content. An alternative type of ubiquitous olfactory display is where the unit sits next to the user. One example of this was developed by Tsaramirsis et al. [14]. They designed and built an olfactory display based on an atomizer to project scent particles into the air. The final device could release up to 24 different scents and was tested with a customized VR game. A smart release algorithm was utilized to make sure that each scent was delivered as ‘clean’ as possible. In total 15 volunteers were recruited to test the effectiveness of the device.

A further approach was proposed by a Japanese research group led by Yanagida [15,16]. In this case, scent delivery was achieved through the use of an air cannon. This type of device delivers scent by shooting out vortex rings right to the user’s nose. The advantage of vortex rings is that the scent inside the vortex remains trapped until the ring loses its shape. Compared to other devices, which produce a constant smell to the surroundings, vortex rings contain only a small amount of scent particles in each ‘shot’. Hence it can achieve good scent projection whilst minimizing the amount of scent. Furthermore, as the vortex ring can travel a relatively long distance before releasing the scent, it can be set up slightly away from the user, making it less onerous. However, in the original publication, there was no information about user testing, nor did they use aromas in their experiments. In addition, the devices presented were highly complex and the level of information about the construction and design of the air cannon was very limited. This makes replication of this work challenging for other researchers. Another group has tried to tackle this problem and developed an olfactory display system called aBio [17]. In their paper, they described a systematic design of their Subwoofer-based olfactory display. Two olfactory displays were designed so that the smoke rings produced would arrive at a certain point inside the room, at the same time. The final device was tested by both gas sensors and a user test of 16 participants to confirm its functionality. However, this approach was found to be limited by the loud noises created by the actuation process. Moreover, they instructed their volunteers to inhale and exhale at specific times, to make sure that their natural breathing would not blow the vortex ring. In most settings, this would not be achievable in normal use.

Therefore, our group has attempted to develop a simpler olfactory display device, which also functions by using vortex rings, with complete design details. We have also undertaken user testing of our olfactory display with multiple users to show the effectiveness of our design. We believe that using vortex rings can solve some of the issues that have affected previous development, such as device noise and have the potential for wider use in a consumer setting.

## 2. Materials and Methods

### 2.1. Preliminary Experiment

Preliminary experiments were carried out to identify the ideal length of the vortex cannon. As the aim of this project was to build a device that emits scent in a rapid and controlled way, a key factor was to maximize delivery length (distance from the user to the vortex source), whilst maintaining the scent-loaded ring structure. Mathematically, the optimal device parameter can be deduced by first calculating the volume of air displaced by the movement of the piston. This is shown below:(1)$$Vslug=\pi \times {\left(\frac{Dtube}{2}\right)}^{2}\times Hpiston\times \frac{1}{3},$$
where *Vslug* is the volume displaced by the movement of piston, *Dtube* is the diameter of the air chamber, and *Hpiston* is the displacement of the solenoid piston by length. This calculates the displaced volume inside the air chamber by pressing the piston. Furthermore, by obtaining an optimal *Hpiston* value, the total volume displaced can be calculated. We then calculate the stroke ratio by using:(2)$$Rstroke=\frac{4\times Vslug}{\pi \times Da^{3}},$$
where *Rstroke* is the stroke ratio and *Da* is the aperture diameter. It has been found that for a stable vortex ring to be generated, the stroke ratio must be smaller than the theoretical threshold value. In this case, *Rstroke* must be between 3.6 and 4.5 [18]. By putting these two values into *Rstroke* and acknowledging the value of *Vslug*, we can successfully calculate the appropriate value for aperture diameter. These two equations are used later in Section 2.5 to help establish appropriate device parameters.

To understand this relationship in an experimental setting, a series of preliminary experiments were undertaken. A 0.5-litre water bottle was cut from the bottom, leaving the top part with the cap. The open end was then covered with a piece of thin rubber. Incense sticks were used to produce smoke inside the water bottle, to make visualisation of the rings easier. Three prototypes were produced with the same aperture size (15 mm), the same bottle diameter (50 mm) and the same material. A Solenoid (MCSMO-0630S12STD, Multicomp Pro, London, UK) was used to hit the rubber membrane, which was controlled using an Arduino Uno microcontroller. The solenoid used in this experiment was also used in subsequent experiments. Three different bottles lengths were investigated, 18.5 cm, 20.0 cm and 21.5 cm. Furthermore, the solenoid was secured 1 cm away from the membrane. The final results were recorded by a camera. These tests concluded that it was feasible to produce scent-loaded vortex rings and gave a rough parameter range for the length of the air cannon.

### 2.2. Scent Generation

Once our preliminary experiments were completed, we developed a more advanced vortex cannon. Our olfactory display device comprised two elements: a scent generation component and a scent projection component. Scent generation is the process of generating scented air particles. This was achieved using a pump, a heating element and a vial containing an essential oil. The essential oils were heated and the air pumped through the vial, which was then loaded with scent headspace. The transported scent then enters the projection side, which comprises an air chamber and a solenoid/membrane actuator. An overview of the system is shown in Figure 1.

The essential oil container was a 20 mL glass vial, as shown in Figure 2e. Tube fittings were put onto the cap and a PTFE tube was placed inside the vials for air to flow. A heater (DB200/2, Techne, London, UK) was then used to heat the vials and increase the diffusing rate of the odour. These scented air particles were then moved from the essential oil jar to the air chamber via a pump, ready for scent projection. The pump (NMS010, KNF, Witney, UK) used in this project was operated at 3.2 VDC. The operating current was measured to be 0.18 A. The advantages of this approach are the low cost and the easily controllable temperature. Furthermore, a thermal heating element produces nearly no noise while operating, which could distract the user.

### 2.3. Scent Projection

From the preliminary tests, described in Section 2.1, a final vortex canon was constructed. Steel was chosen as the main material as it is strong, durable and would reduce the amount of aroma adhering to the inside of the unit. The length was chosen to be 20 cm and the diameter of the tube was 5 cm based on the results from the preliminary experiments. From the projection side of the unit, a 3D-printed cap was constructed in PLA with different apertures. The membrane was secured at the end of the air chamber using a thin layer of nitrile (VMR, Avantor, Lutterworth, UK) with a thickness of approximately 0.12 mm. A 3D printed plastic plate, with a diameter of 18 mm, was secured in the middle of the membrane to minimise damage of the membrane by the solenoid. A testing rig was constructed to adjust the distance between the projection chamber and the other components, as shown in Figure 2a. A schematic of the device is shown in Figure 2b. The front and the back of the device is shown in Figure 2c,d. A CAD design of the solenoid holder is shown in Figure 2f. Three apertures of different diameters were 3D printed, 11 mm, 17 mm and 23.4 mm. The range was chosen based on the aperture size established in the preliminary research. These were then tested to select the one that produced the best performance. A holder for the gas chamber and a case for the solenoid were 3D printed, which included holes for screws so that they could be fixed in place during each shot. Two holes were drilled near the back of the gas chamber facing right angles to the direction of scent travel. One was used as a gas inlet for the scent, whilst the other was used for the connections to a gas sensor placed inside the chamber. This gas sensor was used to monitor the concentration of the odour throughout the experiments.

### 2.4. Control Unit

A control unit was used to drive the three electrical components, the pump to move scent into the chamber, a solenoid to actuate the membrane and a Bosch BME680 gas sensor. These components were controlled by an Arduino Uno (Arduino, Uno, Turin, Italy) board. A voltage regulator (MCP1826, Microchip, Chandler, AZ, USA) was used in the circuit to drive the pump. A 12 V DC power supply (EP-907, Manson, Beijing, China) was used as a power source. The circuit schematic of the control board is shown in Figure 3.

All measurements from the gas sensor were fed back to Arduino using an I^2^C connection. The minimum time for the solenoid to complete a full cycle of movement was found to be 200 ms. As the gas sensor’s maximum measurement frequency was one reading per second, the solenoid’s frequency was set to be larger than this. Hence the Arduino was programmed so that the gas sensor took a reading every second and the solenoid completed a cycle every two seconds. To achieve maximum efficiency of scent movement into the chamber, its volume was calculated, and it was found that it would take the pump 12 s to fully fill the air chamber. Hence the operating time for the pump was set on a cycle of 24 s, with the first 12 s on and the other 12 s off.

### 2.5. Testing of Air Cannon Parameters

To project vortex rings in the most effective and efficient way, mathematical calculations and experiments were undertaken to define the optimal aperture size and hitting force. Three apertures with different diameters (11 mm, 17 mm and 23.4 mm) were 3D printed and tested to identify which generated the farthest-travelled vortex ring. To do this, an incense stick was used to generate smoke to fill the chamber. Throughout the experiment, the ventilation system inside the experimental room was turned on. Since the ventilation system was behind the device, it affected the results slightly as the smoke rings were slowed by the oncoming airflow. The experimental process was video recorded and analysed in Section 3.2.

The hitting force from the solenoid also affects the shape of the vortex ring. As we are using a solenoid in our setup, which is an on/off type device, we can only alter the distance of the solenoid from the membrane. The aperture was kept constant based on earlier experimental results. The pump was turned off throughout the experiments and incense sticks were used. By changing the distance between the solenoid holder and membrane, it effectively changes the displacement of the membrane during ‘shooting’, hence changing the amount of force applied.

Three different distances were marked out. These were measured from the solenoid holder to the chamber and were set to 11 mm, 15 mm and 17 mm. The length of the piston was 17.5 mm. Hence the theoretical displacement of the membrane was 65 mm, 25 mm and 5 mm. The experiments were video recorded with the results analysed in Section 3.2.

### 2.6. Gas Sensor Testing

As described previously, a VOC sensor was placed inside the chamber to monitor the scent particle concentration when the display was in operation. During the experiments, scented air was pumped from the glass vials to the air cannon for a period of 90 s. Three sets of experiments were conducted to confirm the reliability and reproducibility of data. The result of this experiment is shown in Section 3.3.

### 2.7. Volunteer Testing

The final testing was carried out by invited volunteers. The aim of this was to test the user experience of the device. A group of four volunteers were invited with each assigned a 30-min slot. Ethical approval was granted by the University of Warwick (REGO-2019-ENG-021). The procedure for volunteer testing is shown in Table 1. A fume extract unit was placed on the table for ventilation purposes. Overall, three sets of scents were used: Lime Cold Pressed, Cinnamon Bark and Peppermint Arvensis. Each scented essential oil was diluted with two different volumes of ethanol: 50% and 75% mix (as would be used in perfume). This setup is shown in Figure 4. During the test, each participant was exposed to all three different smells at different distances from the unit, with two scents 80 cm away from the unit and one 55 cm. Each smell was repeated with two different concentrations (50% and 75% dilution). Each volunteer experienced 3 smells × 2 concentrations, hence 6 trials. Overall, a total of 24 results were collected. Volunteers were asked to report when they first smelled the odour after the unit “fired” and the observed intensity of smell (ranking from 1 to 10 with 1 being very weak and 10 being very strong). The time when they first noticed the smell was recorded. Volunteers were also asked to complete questionnaires and the results can be seen in Section 3.4.

For calibration purposes, essential oil diluted with 25% ethanol had a VOC measurement of 37–42 KOhms reading (BME680, Bosch, Gerlingen, Germany) and 29–35 ppm reading (Tiger, Ion, Royston, UK), whilst essential oil diluted with 50% ethanol has a VOC measurement of 75–82 KOhms reading and 21–25 ppm reading. Note that the BME680 is a commercial general purpose VOC sensor based on an n-type metal oxide. It detects volatile organic compounds based on a change in resistance of the metal oxide. However, the result given is one resistance value which represents an overall VOC content [19]. Therefore, it is likely that the concentration measurement comprises both the ethanol and the essential oil. However, ethanol has a very high odour detection threshold and is likely to be almost odourless to the user.

The olfactory display unit was placed at a height that was equivalent to the height of a person’s nose when sitting up, which was estimated to be 1.1 m. A heating block was placed inside the extraction unit and the temperature was set to 45 °C. The distance between the olfactory unit and the edge of the fume cupboard was marked out at 25 cm. Two more marks were put on the ground to secure the chair for volunteers to use at distances of 30 cm and 55 cm away from the edge of the fume cupboard. Therefore, every participant sat roughly the same distance away from the unit for each arrangement (close setting = 25 + 30 = 55 cm and further setting = 25 + 55 = 80 cm).

## 3. Results

### 3.1. Preliminary Experiment

For the preliminary experiment, three prototypes with different cannon lengths (18.5 cm, 20 cm, 21.5 cm) were produced. A solenoid was used to strike the bottom of the prototype at a constant rate of one strike per two seconds to produce vortex rings. A black screen was put along the projection direction of the vortex rings to help visualise the rings. A distance reference was implemented on the black screen to show the distance travelled by the vortex rings. The results from this experiment are presented in Figure 5.

From Figure 5, the prototype with a length of 20 cm produced the best result, whereas the prototype with a length of 18.5 cm produced the worst result. This indicates that a small change in unit length strongly influenced projection distance. Figure 5c shows that for a prototype with a length of 21.5 cm, some smoke rings could be formed, but the maximum distance travelled was low. Though this cannot be seen very clearly in Figure 5, the farthest distance the vortex rings could travel for a 20 cm prototype was 50 cm, whereas, for the 21.5 cm setting, the farthest distance was only 10 cm. These results show that implementing a prototype of length 20 cm, diameter 5 cm and aperture size 1.5 cm can produce reliable smoke rings that travel a suitable distance.

### 3.2. Air Cannon Parameters

To implement fine-tuning of the aperture size and hitting force, two further experiments were undertaken. For the aperture size experiment, it showed that the 11 mm diameter aperture produced a cloud of fast-moving smoke. However, it was not a vortex ring and the cloud dissipated quickly. Next, the 23 mm aperture did not produce any vortex rings. There was fluid coming out from the aperture when the solenoid was activated; however, there was not enough energy to push it and keep the fluid moving forward. An aperture with a diameter of 17 mm was tested and a good result was found. For this aperture dimension, a visible smoke ring was produced, and it travelled a relatively long distance before it deformed. These results are presented in Figure 6.

The amount of force applied to the membrane also affects the stability of vortex rings. This was tested by adjusting the displacement between the solenoid and membrane. The results are shown in Figure 7.

When the theoretical displacement of the membrane was 65 mm, the actual travelling distance of the vortex ring, in each shot, cannot be calculated because the solenoid was placed too close to the membrane. Hence the membrane was already stretched before the solenoid was even turned on. There was less momentum created under this condition for the formation of a vortex ring. Vortex rings were formed for a noticeably shorter period of time and then deformed. When the displacement of the membrane was 5 mm, the air cannon produced a small and unstable ring, which was hard to capture on camera. The ring quickly deformed and could only be seen as a cloud of smoke. However, when the displacement of the membrane was 25 mm, the smoke ring was visible and could travel a relatively long distance before it deformed.

Therefore, to make the vortex ring travel the farthest distance, the ideal arrangement is to have an aperture of 17 mm in diameter and the gas chamber placed 15 mm from the solenoid (which produces a membrane displacement of 25 mm). These settings were then used for all further experiments. By putting a *Hpiston* value of 25 mm and an aperture of 17 mm, we could calculate the *Rstroke* ratio based on Formula (2). This gives us a result of 4.24, which is between 3.6 and 4.5. Therefore, our empirical measurements aligned with air cannon theory.

### 3.3. Gas Sensor Measurements

To assess the functionality of the device, three repeated sets of test results were collected, as shown in Figure 8. Vertical lines are drawn at intervals of 12 s, which was a half working cycle for the pump. As discussed before, as this sensor uses an n-type metal-oxide material, a fall in resistance is associated with an increase in scent concentration. The graph suggests that the three test results follow a similar pattern, which concludes that the results are reproducible.

For the first 12 s, the sensor started to take measurements and stabilized its readings. Then from 12 to 24 s, there was an increase in the resistance value, which was associated with the air inside the unit getting cleaner as the pump had stopped running and the solenoid shooting out the scented air. From 24 to 36 s, there was a drop in resistance value where the opposite occurs, and more scented particles were introduced into the gas chamber. This pattern then continued for a 48-s period, which aligned with the unit’s working cycle. This is marked by the vertical black lines in Figure 8.

The solenoid’s action was not captured by the gas sensor as the amount of scented air shot out in each cycle was too small and the process happened too quickly. However, it was able to give an indication of the amount of scent within the chamber per cycle and the amount of scent projected out of the chamber. After each cycle, the resistance value stayed around 400 to 420 kOhms, which suggests enough scent particles were released to maintain a steady cycle. Hence the pump-off period of 12 s was considered reasonable.

### 3.4. Volunteer Testing and Sensing Performance

The final step was device testing with a group of volunteers. The procedure of the volunteer testing is described in Table 1. A full record of the pre- and post-experimental questionnaire answers can be found in the Appendix A.

From the answers in pre-experimental questionnaires, two men and two women were recruited from the age group (18–24). On average, their self-rated ability to smell was 6.25 out of 10 and none of them had smell dysfunction. For the second part of the experiment, we considered the time taken for the participants to detect an odour and then to identify (if possible) the odour. The first of these measurements is shown in Figure 9 below.

On the x-axis of Figure 9, the relevant testing information is encoded. The first two digits of each name represent the reference number of the volunteer, and the last two digits represent the distance between the olfactory device and the volunteer in centimetres (the close and farther setting). The middle letters represent the type of scent used (PP for Peppermint, CI for Cinnamon and LI for Lime).

Overall, 12 sets (each containing 2 trials with different essential oil concentrations) of scents with different intensities were tested by volunteers. From the data collected, 9 out of 12 sets (75%) indicated that users took less time to smell the odour when the odour intensity was higher. Interestingly, the effect of changing distance did not significantly affect the time taken for users to detect the scent. However, it did affect the intensity, which the volunteers rated on a scale from 1 to 10, with 1 being the weakest and 10 being the strongest (though no intensity training was provided to the volunteers). This result is shown in Figure 10.

From Figure 10, in 7 out of 12 tests (58.3%), the user scored intensity increased as the concentration of the essential oil (in the solution) was increased. This poor result might be associated with the duration of the scent being quite short in a projection-type display, which makes it harder for users to rate the intensity. The result could also be affected by the miss rate of the air cannon. This is because the direction and structure of the vortex ring can be easily affected by a sudden change in airflow, e.g., the breath of the user. However, we decided to let participants breathe normally during the experiment to mimic real-life usage scenario. Next, the change in distance of the display device, from 80 cm to 55 cm, did not significantly affect the intensity that the users experienced. It was also found that a period of 2 to 3 min of ventilation was sufficient to remove any smells from the room.

We then compared the results from the integrated VOC sensor with the human panel. In this application, the volunteers are behaving as the sensing/detection technology and typically this is considered the gold standard for odour detection. Here, we considered the response time and the intensity provided by VOC sensor and the human panel. For the VOC sensor, the change in sensor resistance was used for the magnitude, comparing the resistance before and after the chamber was filled with odour, and the time to reach 90% of its final value as the response time. For the human panel, the absolute intensity and the time to detection was used. Figure 11 shows the results of this comparison. As can be seen, the VOC sensor provides a relatively stable output, which only provides limited differentiation to the different odours, with peppermint generally producing the strongest result and lemon the weakest. The response time is likely to be defined by the pump, where the chamber takes longer to fill that the sensor to respond, thus giving almost identical response times. The samples also contained a high ethanol content, which may limit the sensor’s ability to further differentiate the odours. However, the human panel also shows limited capacity to differentiate odours based on response time and intensity. The response time had a window of around 40 s and variance in intensity across the scale. Interestingly, for peppermint, the longer the detection time, the stronger the perceived intensity, while with cinnamon and lime the opposite occurs, with the longer the time to recognition, the lower the intensity. The latter would be more expected as the odour would diffuse over time leading to a lower intensity. However, it indicates that these parameters are affected by how humans perceive odours and that sensitivity is different depending on the odour. An advantage of human sensing is that the odour detection threshold of ethanol is high, meaning that they are not affected as much by the higher ethanol content [20]. This shows the potential importance of smell training before the experiments and also issues that might arise with such systems in a consumer setting, where subjects would not normally have received training.

In the post-experimental questionnaire, volunteers were asked to provide some feedback on the system. On average, the answer to “how comfortable the experience was” achieved a rating of 7 out of 10. This result showed users feel comfortable while using such a display. Some comments also described how the device could be improved. This is shown in Table 2.

Though we were successful in producing a unit that created odour rings and one where we could vary the intensity, there were several limitations to the study. First, the user had to remain in a fairly small defined area for the odour ring to reach their nose. Potentially, face tracking could reduce this problem, but would add expense to the system. In addition, there was a time delay between switching to different odours. In our case, the switch was manual, but even with an automated system, the delay may be unacceptable. This could be resolved by using a small number of scent cannons in parallel to produce a better effect and allow for the mixing of odours in the air. Adding more odours to the test might also help us to gain a better view of the overall performance, according to volunteer 12. Next, the user subjects were not odour trained and therefore, the ability to identify an odour could have been more limited. A formal test to test their olfaction ability might also be useful to adjust the weighting of their answers. However, we allowed our users to breathe normally during experiments to mimic real-life usage scenarios, which is different from previous studies [17]. Finally, in these experiments, we did not add additional stimuli (sight and sound) to the tests. As volunteer 30 suggested, this might affect his overall experience with the test. We shall be looking at this integration in the future.

## 4. Conclusions

The aim of this paper was to develop a simple air cannon-based olfactory display for single-user operation. This was designed from first principles, with testing of each phase of the design. This was followed by user testing of the system to define the length of time to detect an odour, its nature and its intensity. Overall, these tests conclude the ability of our system to deliver scent to users. The concentration of solution appears to affect how fast the user can detect the odour; however, it did not affect the users’ perception of intensity. Furthermore, the unit proved to function best when the user was sitting within 80 cm of the unit. This showed some of the limitations of using humans as a biological sensing system and the importance of smell training. However, this may be an issue with using the device in a consumer setting, where training would be limited. In the future, development of a more sophisticated scent-switching system should be developed and decreasing the loading time to provide a faster and more accurate result. As the sample size presented in this study was small, we would like to expand the number in later studies. We would also like to increase the number of smells used and incorporate the system with other means of media (sight and sound) to enhance the user experience.

## Figures and Tables

**Figure 1 sensors-23-05189-f001:**
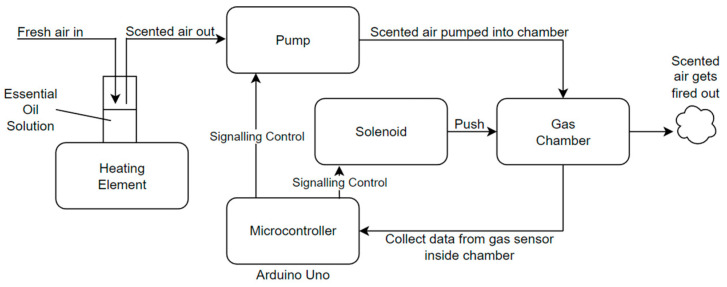
System Overview Diagram.

**Figure 2 sensors-23-05189-f002:**
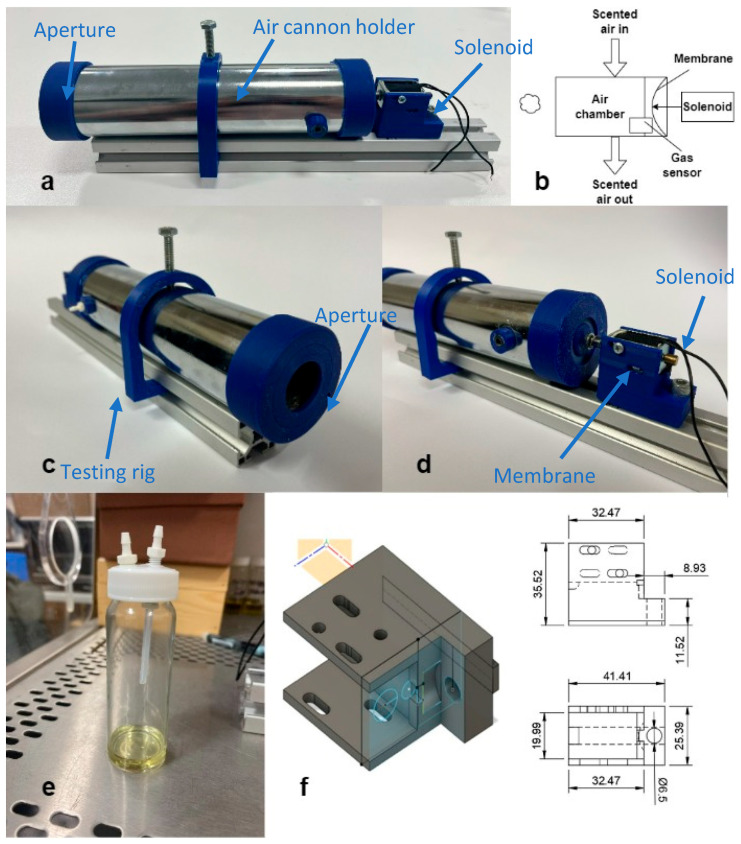
Mechanical design of olfactory display showing device overall view (**a**), device schematic (**b**), front view (**c**), back view (**d**), vial jar (**e**) and CAD solenoid holder (**f**).

**Figure 3 sensors-23-05189-f003:**
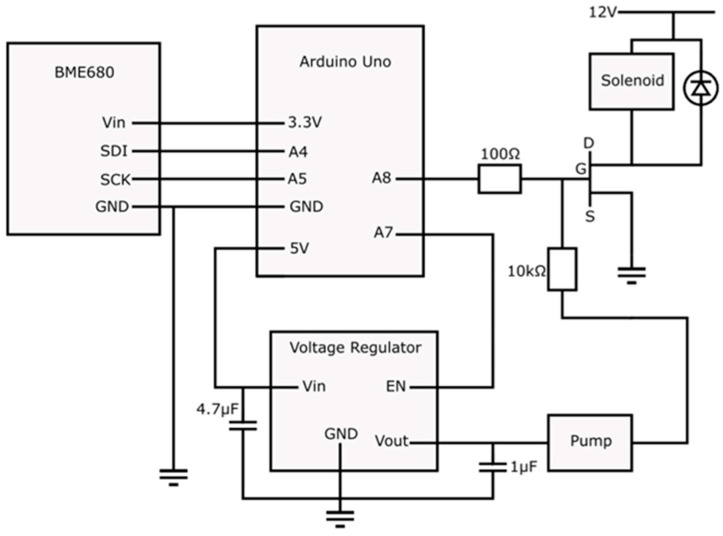
Electrical design of olfactory display.

**Figure 4 sensors-23-05189-f004:**
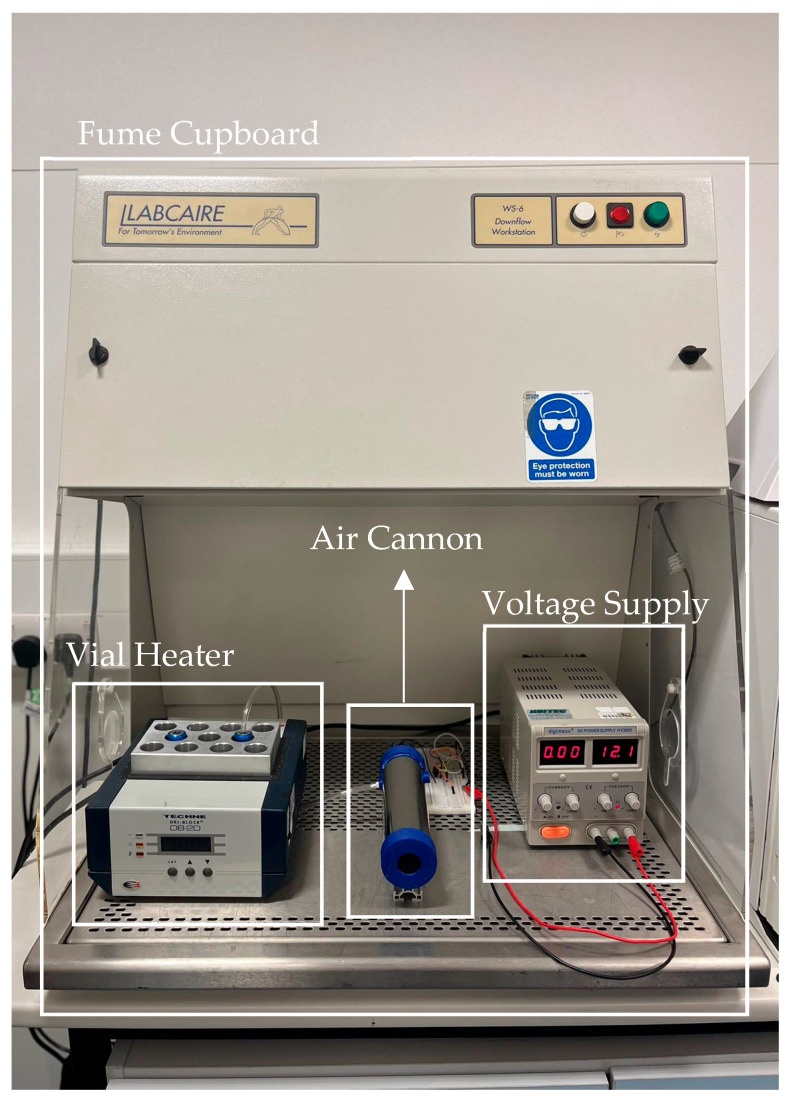
Experiment setup with all sections marked out.

**Figure 5 sensors-23-05189-f005:**
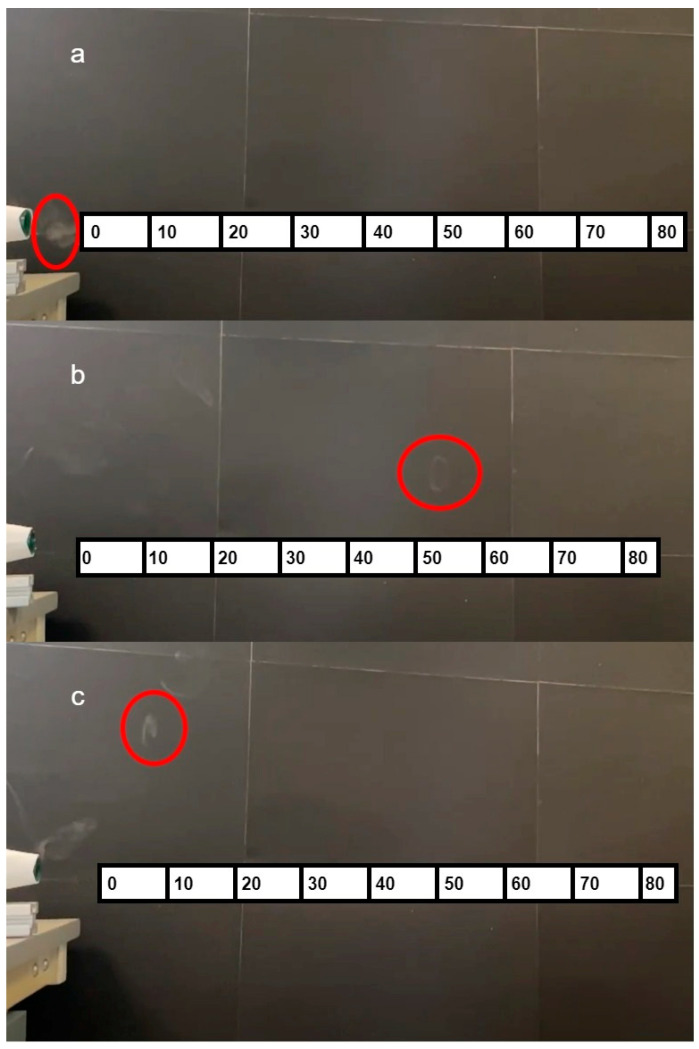
Three prototype results of length with 18.5 cm (**a**), 20 cm (**b**) and 21.5 cm (**c**), farthest ring circled in red, the scale on the graph is measured in centimetres.

**Figure 6 sensors-23-05189-f006:**
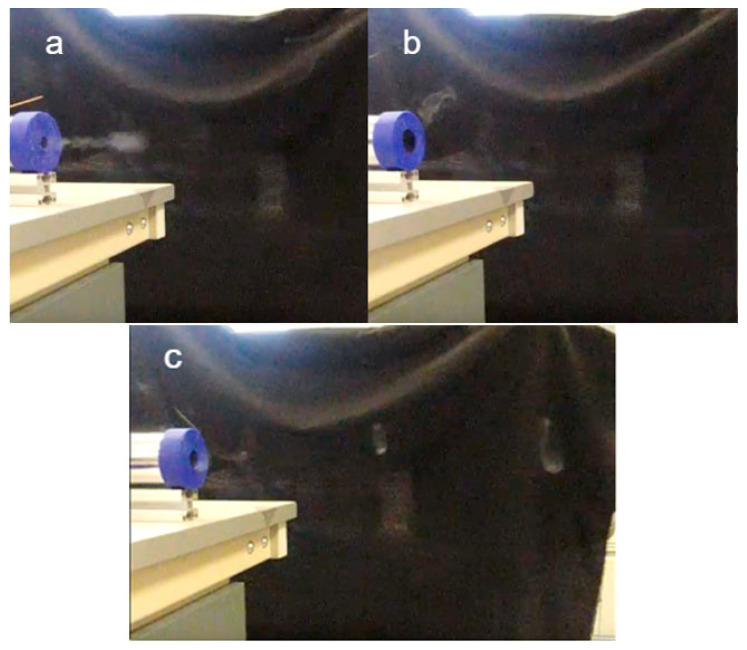
Testing results of aperture with 11 mm (**a**), 23.4 mm (**b**) and 17 mm (**c**).

**Figure 7 sensors-23-05189-f007:**
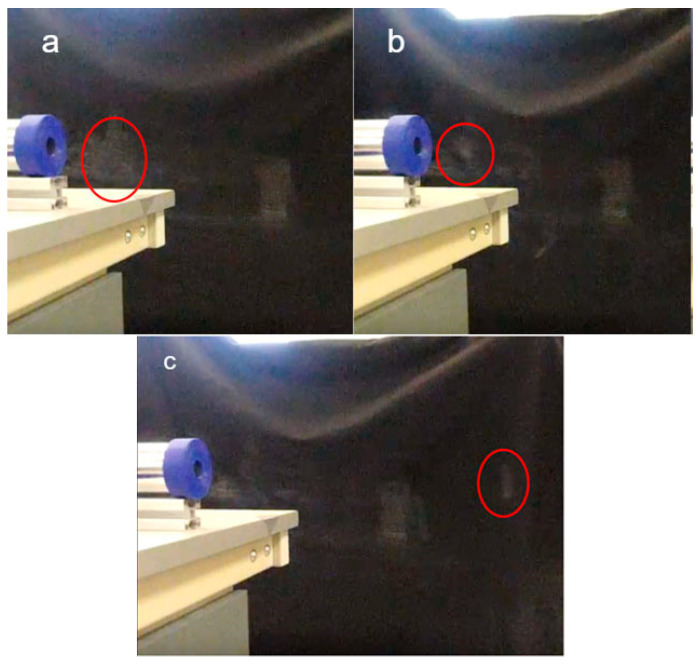
Testing of membrane with theoretical displacement of distances 65 mm (**a**), 5 mm (**b**) and 25 mm (**c**), farthest ring circled in red.

**Figure 8 sensors-23-05189-f008:**
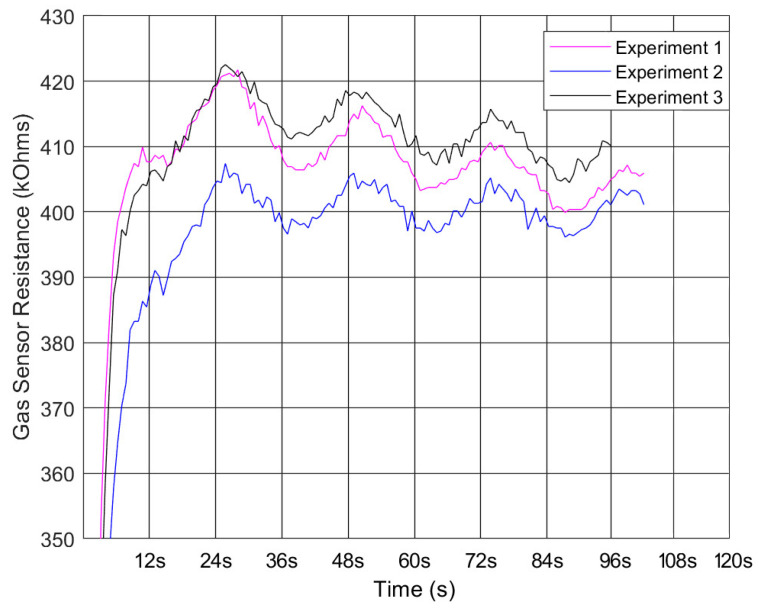
BME680 sensor testing result with three data sets marked with pump on-off periods in vertical lines.

**Figure 9 sensors-23-05189-f009:**
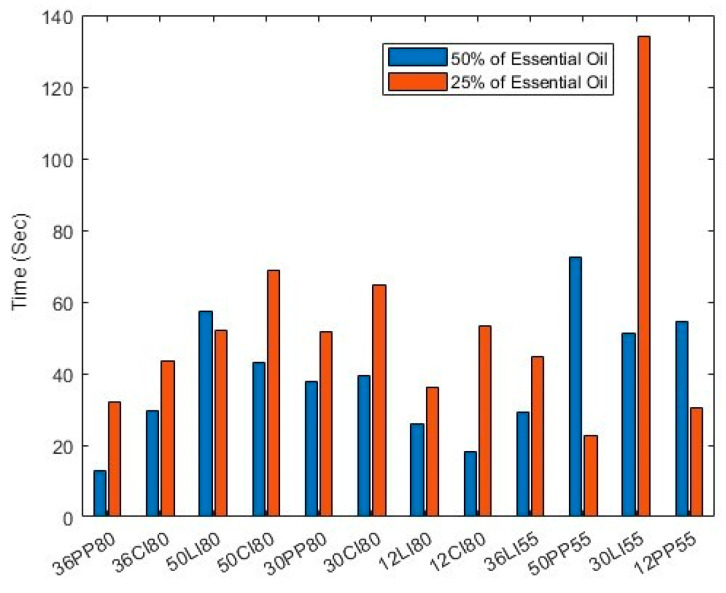
Testing data of the time for the volunteers to smell the odour.

**Figure 10 sensors-23-05189-f010:**
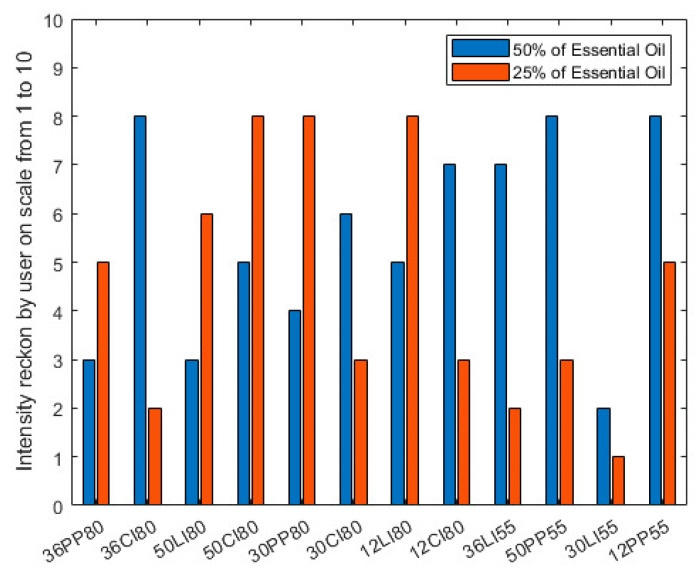
Testing data for the user-rated intensity of odour.

**Figure 11 sensors-23-05189-f011:**
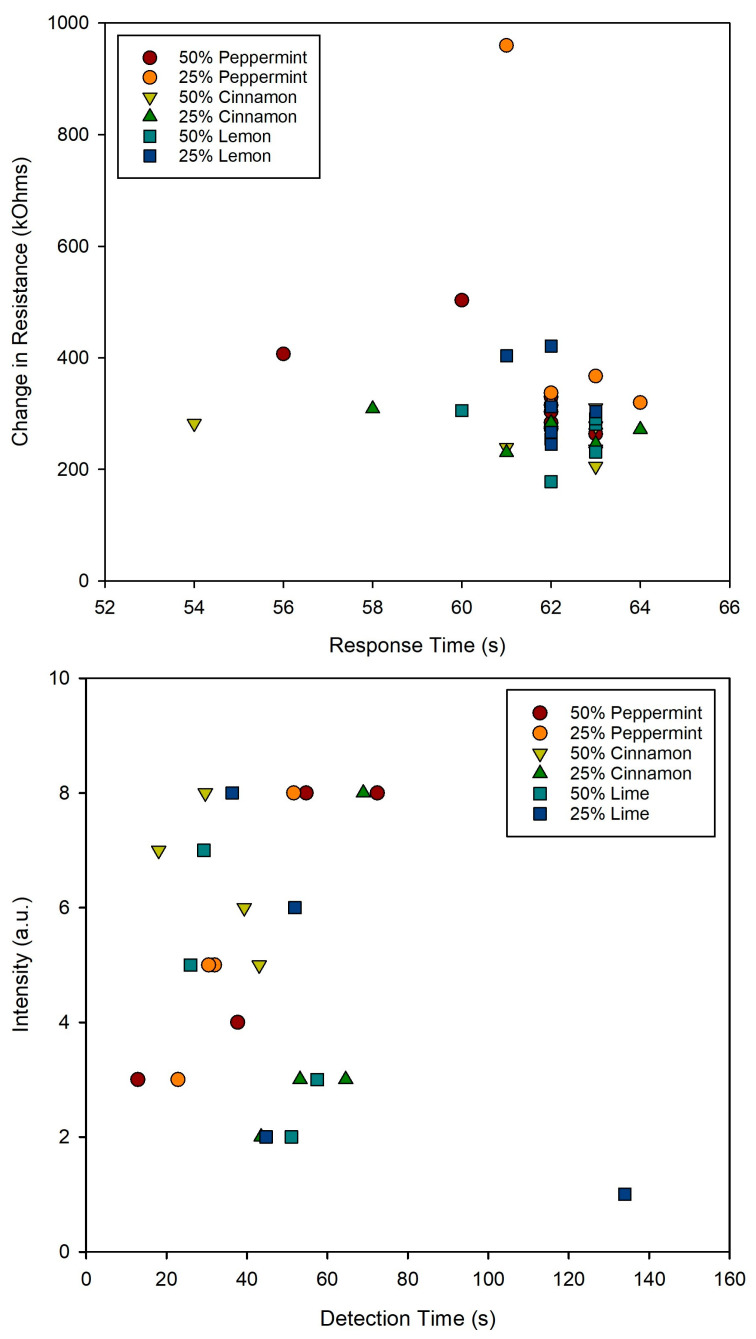
Sensing comparison between the VOC gas sensor and the human volunteers.

**Table 1 sensors-23-05189-t001:** The procedure of volunteer test.

Step	Description
1	The volunteer is given a short pre-experiment questionnaire to provide age range, gender and how good a sense of smell they perceive they have.
2	The volunteer is exposed to different scents. They alert the host when they first smell the scent, give a description of the scent and rate its intensity (from 1 to 10 with 1 being very weak and 10 being very strong).
3	A post-experiment questionnaire is given for volunteers to rate how comfortable they felt about this type of display and provide feedback.

**Table 2 sensors-23-05189-t002:** Volunteer feedback after experiment.

Volunteer	Feedback
12	A few samples smell quite similar, maybe need to change to some different types.
30	If the scents were stronger then it might be easier to guess the smell. If I am in front of the projector then that is all I am concentrating on, rather than the smell.
36	The experiment went well, the scent was a bit hard to guess. Sometimes it could be hard to catch the scent.
50	Having the smell in your surroundings is more immersive than on a mask and feels more realistic.

## Data Availability

Data has been made available in the Appendix A.

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
