# Peer review of "The Development of a Simple Projection-Based, Portable Olfactory Display Device"

_sensors, 2023, doi:10.3390/s23115189_

Round 1

Reviewer 1 Report

The article “The Development of a Simple Projection-based, Portable Olfactory Display Device” describes the generation of flavours in a vortex and some preliminary user experience surveys are conducted. Whilst this reviewer finds the work interesting, some parts are unclear or missing, therefore, this reviewer recommends that a major revision is required before it is suitable for publication.

My comments are below:

·       The authors must add more information on the optimisation process used for the parameters. Some theoretical design equations should be presented to allow readers to engage with the most important parameters in the system.

·       The authors should add a very clear schematic of the internal structure of the device showing the membrane and adjustable components. It is quite hard to understand this in the manuscripts present form.

·       Explanation of the BME680 and what exactly it measures is required. Are receptor materials used how does concentration cause the resistance change etc.

·       Lines 235 and 236 should be “ranking from 1 to 10 with 1 being very weak and 10 being very strong” – Same change for Table 1.

·       Figure quality of Figure 7 needs to be much better for publication, it is not very clear for readers.

·       Is only one ring generated for each length or are multiple rings generated each pumping motion? Figure 8c shows three rings. Are they equally spaced and what are their diameters?

·       The authors should include a schematic of the vortex shape alongside the vortices so that it is visually clear for the reader. Are these shapes independent of the aroma used?

·       The authors should note all the concentrations and estimated number of scent molecules per test. This section still refers to the gas sensor but not much detail is discussed about the measurable concentration. Does the VOC sensor only work for essential oils? Or is the reading calibrated from reference testing?

·       The manuscript structure needs alterations. There are many parts that can be combined. The sections containing Figures 1-5 should be a single figure under one heading of “Vortex inducing aroma generator” as an example, then two sub-headings such as mechanical design and electrical design.

·       More volunteers are required to make any informed decisions about the device, this is necessary. No real conclusion can be made with the data presented in Figure 11 and 12. The reviewer understands the complexity of user surveys but only four volunteers is too small a sample number.

·       Can the authors generate additional types of shapes depending on the parameters used in the system? For example, cylinders, rectangles etc by changing to sine waves, square waves different frequencies etc? If a non-essential oil is used, such as ethanol, does the device still generate vortices or something different?

·       Please check the English on the y-axis for Figure 11. There are still many typos throughout the manuscript. Please double check this for presentation reasons.

Reviewer 2 Report

In this work, the authors developed a simplier air cannon based on olfactory display for single user operation. The testing results demonstrated that their devices could deliver scent to the users. And the main observation was that the concentration of solution mainly affected the response speed of users and had little influence on the user preception of intensity. After consideration, I can not recommand its publication in Sensors because this work had little to do with sensor research. No sensitive materials preparation or sensor device structure design or sensing mechanism were discussed here. It just provided a set-up to produce air cannon. In this view, I think this work may be not suitable to publish by Sensors. 

Round 2

Reviewer 1 Report

The authors have addressed my previous concerns.